# Microendoscopic Lumbar Posterior Decompression Surgery for Lumbar Spinal Stenosis: Literature Review

**DOI:** 10.3390/medicina58030384

**Published:** 2022-03-04

**Authors:** Akinobu Suzuki, Hiroaki Nakamura

**Affiliations:** Department of Orthopaedic Surgery, Osaka City University Graduate School of Medicine, Osaka 545-8585, Japan; hnakamura@med.osaka-cu.ac.jp

**Keywords:** microendoscopic lumbar decompression, lumbar spinal stenosis, lumbar foraminal stenosis

## Abstract

Lumbar spinal stenosis (LSS) is a common disease in the elderly, mostly due to degenerative changes in the lumbar spinal complex. Decompression surgery is the standard surgical treatment for LSS. Classically, total laminectomy—which involves resection of the spinous process, entire laminae and medial facet—has been the standard decompression technique; however, it can cause post-surgical instability. To overcome this disadvantage, various minimally invasive techniques that preserve the stabilization structures of the spine have been developed, and surgeons have begun to re-evaluate decompression surgery from the standpoint of reduced invasiveness and cost. More than two decades have passed since the introduction of microendoscopic spine surgery, and studies continue to shed light on its advantages and limitations as new knowledge becomes available. This article is a narrative review of the available literature, along with authors’ experience, regarding the indications, surgical techniques, clinical outcomes, and limitations/complications of microendoscopic decompression for LSS.

## 1. Introduction

Lumbar spinal stenosis (LSS) is caused by the degeneration of the intervertebral disc, facet joint, and ligamentum flavum. LSS causes low back pain, leg pain, and leg numbness with intermittent claudication, negatively affecting the ability to perform activities of daily living and quality of life in the elderly. The prevalence of LSS is reportedly 11% in the general population and 25–39% in the clinical population, and this prevalence increases with age [1]. Due to rapid aging of the population [2] and accumulating evidence indicating the effectiveness of surgery for lumbar diseases [3,4,5], the number of lumbar surgeries is increasing worldwide [6,7,8,9].

Posterior decompression is a standard surgical procedure for lumbar disease. In 1909, Oppenheim and Krause [10] reported the first results for lumbar laminectomy and discectomy. Classic total laminectomy—which includes the resection of the spinous process, entire laminae, and medial facet—is effective for LSS but can cause post-surgical instability due to the destruction of spinal stabilization structures. However, various less-invasive techniques have been developed to address this disadvantage. One of the cardinal advancements in minimally invasive lumbar surgery has been the introduction of microendoscopic surgery with a tubular retractor by Foley and Smith [11]. This technique was developed to treat lumbar disc herniation, but the system has also been applied for the treatment of LSS.

This article reviews and summarizes the available literature, along with the author’s experiences, on indications, surgical techniques, clinical outcomes, and limitation/complications of microendoscopic decompression with a tubular retractor for LSS.

## 2. Surgical Indication

Microendoscopic decompression is indicated for almost all cases that require decompression surgery, including cases with back and/or leg symptoms due to lumbar spinal canal/foraminal stenosis refractory to conservative treatment. Decompression alone may not be indicated for cases of severe instability that require additional fusion surgery. However, the definition of instability is controversial. Furthermore, studies have indicated that the presence of spondylolisthesis does not affect the clinical outcome [12,13,14] and does not necessarily indicate instability, highlighting the need for careful diagnosis of segmental instability using dynamic radiography. In the author’s institution, positive instability is defined as anterior spondylolisthesis with ≥3 mm anteroposterior translation [15,16] and/or ≥5° segmental kyphosis (flexion/extension) [16], lateral spondylolisthesis with ≥3 mm lateral translation, and ≥3° sagittal segmental motion (standing/supine). For such cases, fusion surgery (posterior/transforaminal/lateral lumbar interbody fusion) is usually indicated. However, if the patients fully understand the possibility of additional fusion surgery, microendoscopic decompression can be applied. Microendoscopic decompression may not be applied in cases with a high possibility of intraoperative dural tear, such as in patients with previous decompression surgery, massive ossification of the yellow ligament and facet cysts, as it is difficult to repair massive dural tears with sutures using a microendoscope. However, these are not contraindications to microendoscopic decompression, as skilled surgeons are required for microendoscopic techniques. Murata et al. [17] reported the clinical results of microendoscopic decompression for symptomatic LSS or lumbar foraminal stenosis caused by facet cysts. Dural tear occurred in 4 of 36 cases (11%) with the conventional technique, but no tears were noted among the 12 cases in which preoperative cysts were identified via an injection of indigo carmine into the facet joint. These techniques may be useful for avoiding incidental dural tears.

## 3. Surgical Equipment, Patient Positioning, and Room Setup

Microendoscopic surgery requires specialized equipment. The most popular system is the METRx^®^ system (Medtronic, Memphis, TN, USA), which includes a serial tubular dilator, tubular retractor, and flexible arm assembly to secure the retractor to the table (Figure 1a,b). The system includes 16-mm and 18-mm-diameter tubular retractors for endoscopic surgery, which can be performed using two different dilator lengths for each diameter and a standard or short endoscope. A short endoscope is preferred because a short endoscope and retractor allow the insertion of instruments more obliquely (approximately 9° and 13° maximum between the retractor and instruments for standard and short retractors, respectively). The endoscope provides a 25° oblique view, and a light cable is attached to the endoscope. A disposable attachment with an air suction nozzle is necessary to connect the endoscope and the retractor. In Japan, the SYNCHA^®^ system (Kisco, Kobe, Japan) was developed by Dr. Yoshida. It has a ball link between the retractor and connector to the flexible arm, and the ball link allow for control of the retractor during surgery using a joystick (Figure 1c). In this system, the scope attachment has a nozzle for irrigation in addition to air suction to wash out the surface of the endoscope (Figure 1d), and there are two retractor lengths (50 and 70 mm), although there is only one scope length. 

Camera and video systems are not included in either system, and must be provided separately. The resolution of the camera and video system has evolved into high definition or 4K, which enables safer surgery with a clearer view. 

Longer instruments than used in conventional open surgery are also necessary when performing microendoscopic surgery. With a 25° viewing angle and an ability to rotate, the endoscope provides a much wider visual field than the inside of the retractor. Therefore, curved Kerrison rongeurs and curved high-speed drills are important for the manipulation of the contralateral side in the paramedian approach (Figure 1e,f). 

An example of room setup is shown in Figure 2. The patient is positioned with abdominal decompression to avoid intraoperative venous bleeding. The authors use a Hall frame with four padded supports, although any other device that decreases abdominal pressure (bow frame, Jackson table, etc.) can be used. A fluoroscope should be used to confirm the operative spinal level. Therefore, the table and flame should be compatible with fluoroscopic imaging. The video tower is placed on the opposite side of the operator. While the flexible arm assembly can be fixed to either side of the table, the authors prefer to fix it on the opposite side because the arm itself sometimes obstructs the delivery of the equipment from the nurse/surgical assistant.

## 4. Surgical Technique for Spinal Canal Stenosis

### 4.1. Paramedian Approach 

The most common approach for microendoscopic decompression is the paramedian (unilateral) approach (Figure 3a) [18,19]. A skin incision is made at the paramedian area (approximately 5 mm lateral to the spinous process). The size of the incision should be 1–2 mm longer than the diameter of the tubular retractor to avoid skin ischemia. After the incision of the lumbar fascia, serial tubal dilators are inserted, and a tubular retractor is placed. In this approach, the attachment of the multifidus muscle to the spinous process is not cut. However, some surgeons may prefer the conventional approach of cutting the multifidus to avoid muscle invasion into the retractor. The endoscope is attached to the tubular retractor, and the residual muscle and soft tissues on the lamina and facet joint are removed with adequate haemostasis using bipolar cautery (Figure 4a). Laminotomy is initiated from the base of upper spinous process with a high-speed drill to secure the contralateral surgical field (Figure 4b,c). This is also because the ligamentum flavum covers the dural tube on the cranial side. Thereafter, medial facetectomy and detachment of the ligamentum flavum are achieved using a high-speed drill, curette, and Kerrison rongeurs (Figure 4d–l). Decompression can be initiated from the approach side (Figure 4b–e,l) or contralateral side (Figure 4f–k), but the removal of the ligamentum flavum should occur in the late stage because it protects the dural tube and nerve root from incidental injury. In this approach, the facet joint of the contralateral side can be easily preserved, whereas excessive resection of the facet can occur on the approach side [20,21]. Therefore, surgeons must be conscious of “trumpet facetectomy”, especially on the approach side. During decompression of the contralateral side, the dissection between the laminar and flavum aids in the identification of the direction for contralateral decompression (Figure 4g). The ligamentum flavum is removed after detachment from the laminae, and additional medial facetectomy is performed if necessary.

### 4.2. Midline Approach 

Reports have described two types of midline approach (Figure 3b). Yagi et al. [22] described a midline approach involving osteotomy of the upper spinous process. In their technique, spinous process osteotomy is performed after sequential dilation of the multifidus muscle using a tubal dilator and retractor; the tubular retractor is then moved into the centre. The authors noted that this approach can preserve the attachment of multifidus muscle and reported a case of spontaneous union at the base of the spinous process. However, they did not report the union rate for the spinous process. In the midline approach described by Mikami et al. [23], the caudal part of the upper spinous process is excavated using a high-speed drill while preserving the periosteum; the tubular retractor is then placed between the spinous processes. The authors noted that the supraspinous ligament and shallow layer of the interspinous ligament are almost entirely preserved because the cut line is parallel to the fibres, and the periosteum of the spinous process is preserved. However, the bone tissue of the spinous process partially disappears in this technique, and the exact influence has not been elucidated. These two midline approaches, relative to the paramedian approach, allow the preservation of the facet joint and are well indicated in cases with narrow laminar width such as those in the upper lumbar level.

## 5. Surgical Technique for Foraminal Stenosis

Microendoscopic decompression is also effective for the treatment of foraminal stenosis [24]. A skin incision is made at approximately 2 cm lateral to the lateral edge of the pedicles (5–10 cm lateral to the midline). After incision of the lumbar fascia, serial tubal dilators are inserted into the longissimus muscle (Figure 5a). Thereafter, the tubular retractor is placed just lateral to the facet joint (superior articular process of the lower vertebrae) on the upper and lower transverse processes. The caudal border of the transverse process of the upper vertebrae, the cranial border of the transverse process of the lower vertebrae/sacral ala, and the lateral part of the superior articular process of the lower vertebrae are drilled out until the transverse/lumbosacral ligament is released (Figure 5b). The transverse/lumbosacral ligament is resected, and the nerve root is identified. The identification of the nerve root is easier in the medial part than in the lateral part of the surgical field because the nerve root runs anteriorly in the lateral part. Additional resection of the caudal part of the upper pedicle is necessary for patients with up-down stenosis (Figure 5b), and resection of the ligamentum flavum and/or cranial part of the superior articular process of the lower vertebra may be necessary for those with front-back stenosis at the foramen [25]. However, a biomechanical study revealed that, to prevent postoperative instability, no more than 50% of the lateral part of the facet and/or pars interarticularis should be removed [26]. Studies regarding L5/S foraminal stenosis have reported that most stenoses exist outside the centre of the pedicle, and that 36% of them are extraforaminal stenoses [27]. In the extraforaminal area of L5/S1, the nerve root can be entrapped between the sacral ala and osteophyte originating from L5 or sacrum at the disc level [28]. Therefore, adequate decompression via partial resection of the sacral ala should be carefully confirmed in the foraminal stenosis of L5/S.

## 6. Learning Curve

All surgeons experience a learning curve while acquiring skills for any procedure. The endoscope provides a magnified and illuminated view of the surgical field with minimal incision, whereas it is hard for the assistant to aid the surgeon because of the narrow access. Nowitzke [29] demonstrated that the learning curve for microendoscopic discectomy was approximately 30 cases. Nomura et al. [30] investigated the learning curve for microendoscopic decompression surgery for LSS and reported that the intraoperative blood loss stabilized after the first 30 cases. The operating time decreased along a natural logarithmic function over the span of 480 cases, but the graph showed a rapid decrease over the first 30 cases. Interestingly, they also reported that perioperative complications (mainly dural tear) occurred at any time even after mastery of the technique. Preclinical simulation with a bone model may help surgeons overcome the learning curve more quickly and safely, although simple cases should be selected for the first 30 cases.

## 7. Clinical Outcomes of Microendoscopic Lumbar Decompression

### 7.1. Spinal Canal Stenosis

In 2002, Palmer et al. [31] first reported clinical outcomes in eight patients who underwent microendoscopic decompression for LSS with spondylolisthesis. Although it was a small case series with a short follow-up (4–7 months), patients experienced a decrease in visual analogue scale (VAS) scores for pain after surgery, and the questionnaire indicate that good outcomes were achieved, ranging from 63% to 88%. Palmer et al. [32] also reported the surgical outcomes of 17 patients with LSS in 2002, but the report did not include an evaluation of the patients’ symptoms. In the same year, Khoo and Fessler [33] reported the short-term outcomes of 25 patients treated with microendoscopic decompression for LSS. The results showed improvements in leg pain and low back pain in 88% and 72% of the patients, respectively, at 1 year. Following these reports, many studies have described successful functional improvements in microendoscopic decompression for LSS. Ikuta et al. [20] retrospectively evaluated 47 cases of microendoscopic decompression for LSS without spondylolisthesis, and the recovery rate of the Japanese Orthopaedic Association (JOA) score for low back pain was 72% at the final follow-up (mean 22 months). Subsequently, long-term outcomes have been reported by several authors. Asgarzadie and Khoo [21] reported midterm outcomes, noting that the Oswestry Disability Index (ODI) and 36-Item Short Form Survey (SF-36) scores improved from preoperative values of 46 and 2.2 points, respectively, to 26 and 2.8 points, respectively, at 3 years. Castro-Menendez et al. [34] analysed the prospectively collected data of 50 patients that underwent microendoscopic bilateral decompression via the paramedian approach. They reported mean decreases in ODI, VAS scores for leg pain, and VAS scores for lumbar pain of 30.23, 6.02, and 0.84, respectively, at the final review (mean: 48 months). Aihara et al. [35] reported the ≥10-year outcomes of patients that underwent microendoscopic decompression for LSS. In 46 patients without spondylolisthesis, the mean decrease in the VAS scores for low back pain, leg pain, and leg numbness were 16.9, 37.9, and 30.9, respectively, at the final review (mean: 138.4 months). Their findings also revealed that scores for all categories of the Japanese Orthopaedic Association Back Pain Evaluation Questionnaire (JOABPEQ) score, including those for “pain-related disorders”, “lumbar spine dysfunction”, “gait disturbance”, “social life disturbance”, and “psychological disorders,” significantly improved after surgery. Gupta et al. [36] reported long-term functional outcomes of 953 cases of endoscopic decompression for LSS. The reported Prolo scores were excellent in 89.85%, good in 1.59%, and poor in 8.55% of patients. They also showed that repeat decompression surgery was required at the same level in 1.68% and at different levels in 2.2% of patients, whereas no patients required fusion surgery. Overall, previous reports have demonstrated that the clinical outcomes of microendoscopic lumbar decompression for LSS are good in both the short- and long-term period after surgery.

### 7.2. Spinal Canal Stenosis with Spondylolisthesis

It remains unclear whether decompression surgery, rather than fusion surgery, should be employed for spinal stenosis with spondylolisthesis. As noted above, Palmer et al. [31] were the first to apply microendoscopic decompression for LSS in patients with degenerative spondylolisthesis. Several studies have also shown the efficacy of microendoscopic decompression for spondylolisthesis. Ikuta et al. [37] reported the clinical outcomes of 37 cases of spondylolisthesis, and the mean recovery rate of the JOA score was 64% at the final follow-up (mean: 38 months). The authors reported that there was no significant difference between the preoperative and postoperative values for dynamic sagittal angle and percentage slip, although one patient (2.7%) required fusion surgery due to postoperative instability. Minamide et al. conducted several studies focusing on the clinical outcomes of lumbar degenerative spondylolisthesis [12,13,15]. First, they compared the 5-year clinical outcomes of microendoscopic decompression between patients with (*n* = 61) and without (*n* = 71) degenerative spondylolisthesis [12]. Their results showed that clinical outcomes including JOA score, Roland-Morris Disability Questionnaire (RDQ), and SF-36 were similar in the two groups. Furthermore, the rate of progressive spinal instability after surgery and/or additional surgery was also similar between the groups. In their second study [13], they compared clinical outcomes between patients with (*n* = 86) and without (*n* = 156) advanced degenerative spondylolisthesis (percentage slip ≥ 20, dynamic translation ≥ 5%, or local kyphosis ≥ 5°), reporting no significant differences in JOA score, VAS, RDQ, or SF-36 at the final follow-up. Additional fusion surgery was required in 5.1% of patients in the non-advanced spondylolisthesis group and 6.9% of patients in the advanced spondylolisthesis group, whereas additional decompression surgery was required in 3.2% of patients in the non-advanced spondylolisthesis group and 6.9% patients in the advanced spondylolisthesis group. In their third study [15], they reported the results of a subgroup analysis in which spondylolisthesis was divided into three stages: early-stage (disc height loss < 1/3, percentage slip < 10%), advanced-stage (disc height loss ≤ 2/3, percentage slip ≥ 10%, dynamic translation ≥ 3 mm), and end-stage (disc height loss > 2/3, dynamic translation < 3 mm). The recovery rate of the JOA score was significantly lower in the advanced-stage group than in the early- and end-stage groups, and the rate of additional decompression or fusion surgery was also higher in the advanced-stage group. Other authors have also compared clinical outcomes for LSS in patients with and without spondylolisthesis. Kobayashi et al. [16] compared clinical outcomes 5 years after microscopic or microendoscopic posterior decompression for LSS with and without degenerative spondylolisthesis. They reported no significant differences in improvements in low back pain, JOA score, or reoperation rate between the groups. Aihara et al. [35] compared long-term outcomes between patients with (percentage slip ≥ 5%) and without spondylolisthesis. In their study, there was no significant difference in the degree of improvement in JOABPEQ scores or VAS scores for low back pain, leg pain, or leg numbness between the groups. The reoperation rates at the final follow-up over 10 years were 17.4% in the spondylolisthesis group and 17.6% in the non-spondylolisthesis group. Based on these studies, the clinical outcomes of microendoscopic decompression are similar for LSS with and without spondylolisthesis.

### 7.3. Complications

Several complications have been reported in patients undergoing microendoscopic lumbar decompression for LSS, the most frequent of which is dural tear. Massive dural tears require suturing under a microendoscope or microscope with additional incision, and pin-hole tears are usually repaired with fibrin glue, subcutaneous fat, and/or artificial materials such as polyglycol membrane. In the literature, which includes cases in the early period, the incidence of dural tear ranges from 10–16% [21,31,32,33]. Advancements in surgical skill and equipment have decreased this rate to 1.2–8.5% [13,20,35,38,39]. Tsutsumimoto et al. [38] prospectively investigated the incidence of dural tears and risk factors on clinical outcomes in 555 consecutive patients. The incidence of dural tear was 5.05%, and the risk factors for dural tear included the patient’s age and bilateral decompression via a unilateral approach. Comparisons with an age-, sex-, and procedure-matched control group showed that the recovery rate for the JOA score at 6 months after surgery was significantly lower in those with dural tears than in those without, whereas the ODI was similar between the groups. Soma et al. [40] investigated the incidence of dural tear in microendoscopic discectomy, microendoscopic laminotomy, and microendoscopic laminotomy with interbody fusion, reporting incidence rates of 3.0%, 8.1%, and 7.3%, respectively. Their findings indicated that clinical outcomes, including numerical rating scale score, ODI, JOA score, and SF-36, were similar between the dural tear group and matched controls. Some studies have also examined data related to accidental dural tearing in various types of lumbar surgery [41,42,43]. However, it remains unclear whether dural tear negatively impacts clinical outcomes. Avoiding dural tear is preferable, but adequate repair without damage of the cauda equina by aspiration is also important to avoid the negative impact of dural tear.

Epidural hematoma is also an important complication. Post-surgical bleeding after microendoscopic surgery cannot be avoided due to the small dead space, and it can cause neurological deterioration. Generally, the incidence of severe symptomatic epidural hematoma does not exceed 5.0% [13,20,34,44]. However, this rate markedly increases when asymptomatic hematoma is included. Ikuta et al. [45] conducted a magnetic resonance imaging (MRI) study after microendoscopic surgery, observing epidural hematoma in 33% of patients 1 week after surgery. Patients with epidural hematoma at 1 week exhibited less expansion of the dural sac after 1 year, and the RDQ and Prolo scale scores were significantly worse than those in patients without hematoma at 1 week. Merter and Shibayama [46] also conducted an MRI study to investigate the appropriate location of the drain output in microendoscopic surgery. They divided the patients into three groups according to the location of the drain output: (1) within the incision, (2) 1 cm lateral to the incision, and (3) 5 cm lateral to the incision. Although they did not evaluate the patients’ symptoms, the 5- group experienced a significantly larger hematoma volume and a smaller cross-sectional area of the dural sac at 24 h after surgery than the other groups. The authors indicated that this may have been due to exit loss given the large-angle bend of the tube and recommended locating the drain output within or close to the incision without any angulation. In addition to careful haemostasis with bone wax and bipolar cautery, adequate insertion and management of drain tube may be important for preventing epidural hematoma.

Other reported complications include nerve root injury/transient neuralgia (0.42–10.5%) [13,18,36,39,47], facet fracture/resection (2.6–6.4%) [20,36,47], surgical site infection (0.4–4.0%) [13,34,44], and wrong level of surgery (0.3–3.3%) [18,36,39]. Fourtney et al. [48] conducted a systematic review and compared the incidence of complications between minimal access tubular-assisted spine surgery and traditional open surgery. They concluded that minimal access tubular-assisted spine surgery does not decrease the rate of complications in patients undergoing posterior lumbar spinal decompression or fusion. As described above, microendoscopic surgery requires a considerable learning curve. In their paper focusing on the complications of microendoscopic procedures, Ikuta et al. [47] stated that “Most of the complications occurred in the initial series of patients, and the incidence of complications decreased with an increase in the surgeon’s experience and the application of several preventive measures against the complications.” Surgeons should know where they are on the learning curve, what kind of complications can occur, and the various surgical techniques that can be used to prevent or manage complications. 

### 7.4. Comparison with Other Surgical Techniques

The clinical outcomes of microendoscopic decompression have been compared with those of other surgical methods in several studies (Table 1). Khoo et al. [33] reported that microendoscopic decompression offered similar short-term outcomes with a significant reduction in operative blood loss, hospital stay, and use of narcotics when compared with conventional open laminectomy. Rahman et al. [49] also reported that cases treated with microendoscopic decompression had a shorter operating time, less blood loss, shorter hospital stay, and lower complication rates. Khoo and Rahman used the paramedian approach in their studies, while Yagi et al. [22] compared cases of microendoscopic decompression via the midline approach and open laminectomy. The authors reported not only less blood loss, shorter hospital stay, and less use of analgesics for the microendoscopic approach, but also lower serum creatine phosphokinase levels at 24 h after surgery. The microendoscopic decompression group also exhibit lower VAS scores for low back pain at every time point up to 1 year than the conventional open laminectomy group. These reports clearly highlight the reduced invasiveness of microendoscopic decompression when compared with conventional open laminectomy.

Clinical outcomes of microendoscopic decompression have also been compared with those of other less invasive techniques. Ikuta et al. [20] compared clinical outcomes between microendoscopic decompression and microscopic decompression, reporting less blood loss, shorter hospital stays, and less use of analgesics in the microendoscopic decompression group. The improvements in JOA score and VAS scores for low back pain and leg pain were similar between the groups, but the complication rate was significantly higher in the microendoscopic decompression group (25% vs. 14%). Fujimoto et al. [50] compared microendoscopic decompression with microscopic decompression and reported shorter operating time, less blood loss, shorter hospital stays, lower serum C-reactive protein (CRP) level, and lower NSAID dose in the microendoscopic decompression group despite similar clinical outcomes for JOA score and VAS scores for leg pain. Fukushi et al. [51] compared clinical outcomes between microendoscopic decompression via the midline approach and spinous process-splitting laminectomy, which is an open but minimally invasive technique. The groups experienced similar clinical outcomes, but the serum CRP levels at 3 and 7 days after surgery were lower in the microendoscopic decompression group. Thus, microendoscopic decompression seems less invasive than open or microscopic minimally invasive techniques.

Recently, percutaneous endoscopic surgery with saline irrigation has been applied to the treatment of LSS. There are two types of surgery in this category: uniportal full-endoscopic surgery using the endoscope, which includes a working channel in the scope system, and biportal endoscopic surgery with two different skin incisions for the endoscope and for inserting instruments as in arthroscopic knee surgery. Wu et al. [52] compared the clinical outcomes of microendoscopic decompression and uniportal full-endoscopic decompression and reported that the operating time was shorter, but skin incision was longer, for microendoscopic decompression. Improvements in VAS scores for leg pain were similar in the two groups. VAS scores for low back pain and ODI values were worse only at 1 week in the microendoscopic decompression group, but they were similar in the two groups at 6 months and at the final follow-up. Iwai et al. [53] also compared microendoscopic decompression and uniportal full-endoscopic decompression. They reported that the operating time was shorter, but that the hospital stay was longer, for microendoscopic decompression. Similar improvements in numerical rating scale scores for symptoms were observed in the two groups. Ito et al. [54] compared clinical outcomes between microendoscopic decompression and biportal percutaneous endoscopic decompression. In their study, VAS scores for low back pain and leg pain, ODI values, and EuroQol 5-Dimension questionnaire scores improved similarly in both groups at 6 months after surgery. Aygun et al. [55] conducted a randomized trial of microendoscopic decompression and biportal percutaneous endoscopic decompression, which revealed a shorter hospital stay, shorter operating time, and less blood loss for biportal percutaneous endoscopic decompression. Furthermore, clinical outcomes based on ODI, the Zurich Claudication Questionnaire (ZCQ), and the modified MacNab criteria were superior for biportal percutaneous endoscopic decompression. Although these percutaneous endoscopic surgeries require a long learning curve, they seem much less invasive. Further accumulation of evidence may clarify the advantages of microendoscopic and percutaneous endoscopic surgery.

Several studies have compared the clinical results of microendoscopic decompression and fusion surgery for cases involving spondylolisthesis. Hayashi et al. [56] compared microendoscopic decompression and posterior lumbar interbody fusion (PLIF) with cortical bone trajectory pedicle screw insertion. Improvements in JOA score; VAS scores for low back pain, leg pain, and leg numbness; and reoperation rates were similar between the groups. However, blood loss, analgesic use, and serum CRP levels were significantly lower for microendoscopic decompression. Aihara et al. [14] compared microendoscopic decompression and PLIF or posterolateral fusion surgery. They reported a shorter operating time, less blood loss, shorter hospital stays, and greater improvement in the social function domain of the JOABPEQ in the microendoscopic decompression group. Kimura et al. [57] compared 37 cases of microendoscopic decompression via the midline approach and 78 cases of PLIF. Operating time and blood loss were reduced in the microendoscopic decompression group, although clinical outcomes based on JOA, JOABPEQ, ZCQ, and VAS scores were similar between the groups. A recent meta-analysis comparing the effectiveness of decompression surgery and fusion surgery for spondylolisthesis did not reveal the superiority of fusion surgery [58,59]. It is known that a certain percentage of patients with spondylolisthesis will require fusion surgery. However, this percentage is not high, and evidence indicates that microendoscopic decompression can be utilized in most patients with degenerative spondylolisthesis.

### 7.5. Foraminal/Extraforaminal Stenosis

Microendoscopic decompression is also effective for foraminal/extraforaminal stenosis. Because of the deep location, the magnified oblique view of the microscope is a notable advantage for decompression. Several reports have described the clinical outcomes of microendoscopic decompression for foraminal/extraforaminal stenosis (Table 2), but most of the cases in the literature have focused on extraforaminal stenosis at L5/S1. Matsumoto et al. [28] first reported three cases of microendoscopic decompression for extraforaminal entrapment of the L5 spinal nerve at L5/S1. They performed decompression by partially resecting the lateral aspect of the L5/S1 facet, L5 transverse process, and sacral ala but not the osteophyte of the vertebral body. The average recovery rate of the JOA score was 42.7% at the final follow-up (mean: 31 months). Zhou et al. [60] also reported the clinical outcomes of five cases of extraforaminal stenosis at L5/S1 treated with microendoscopic decompression, noting that satisfactory relief from leg pain was obtained in all five cases. In another study, Matsumoto et al. [61] reported the clinical outcomes of 28 cases of microendoscopic (*n* = 19) or microscopic (*n* = 9) decompression for extraforaminal stenosis at L5/S1. Microendoscopic surgery yielded a better (68.6%) average recovery rate of the JOA score, although four cases (14.3%) required fusion surgery at an average of 19.5 months. Yamada et al. [24] also reported clinical outcomes for 32 cases of extraforaminal stenosis at L5/S1 treated via microendoscopic decompression. They reported adding partial (inferior half) pedicle resection in their cases, and the average recovery rate of the JOA score was 60.2% at the final follow-up (mean: 37.4 months, minimum: 2 years). Revision surgery was required in two cases (6.3%) because of residual stenosis in the foramen. Yoshimoto et al. [25] reported midterm clinical outcomes, with an average follow-up of 66.3 months. Their reports included two cases each of L4/5 and L5/6 foraminal stenosis in addition to 16 cases of L5/S1 foraminal stenosis, and the mean recovery rate of the JOA score was 63.9% at the final follow-up. Five cases (25%) required additional surgery during the follow-up period; two cases with recurrent foraminal stenosis at the same level required fusion surgery. Murata et al. [27] reported the largest case series of lumbosacral foraminal stenosis (*n* = 78) to date. The average recovery rate of the JOA score was 56.0% at 2 years, and the success rate (recovery rate of JOA score >25%) was 94.9%. They classified the location of stenosis into three categories (medial foraminal stenosis, lateral foraminal stenosis, and extraforaminal stenosis) and investigated the location of the narrowest part of the stenosis in each group using 3D image fusion with MRI/computed tomography. The results showed that the narrowest part was the lateral foramen in 58% of case and the extra-foramen in 36% cases, with the medial foramen accounting for only 6% of cases. Their finding suggests that most areas of stenosis exist outside the centre of the pedicle, supporting the efficacy and importance of lateral fenestration for lumbosacral foraminal stenosis. 

## 8. Conclusions

Since the introduction of microendoscopic spine surgery more than 20 years ago, many studies have demonstrated the safety and efficacy of microendoscopic lumbar decompression surgery for LSS, degenerative spondylolisthesis, and foraminal stenosis. The technology for minimizing invasiveness and improving safety has continued to advance, and novel techniques and surgical equipment are being introduced. However, it is important to continue investigating the efficacy/safety of these modalities and accumulating scientific evidence to improve patient outcomes.

## Figures and Tables

**Figure 1 medicina-58-00384-f001:**
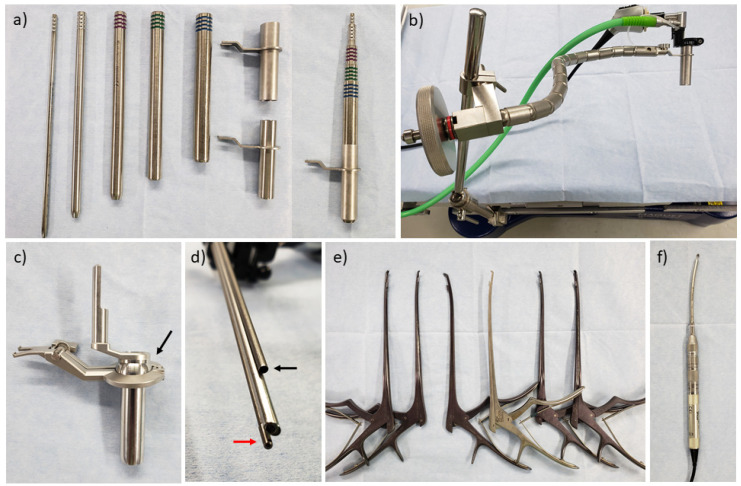
Surgical equipment for tubular microendoscopic decompression surgery. (**a**) Serial tubular dilator and retractor (METRx^®^). (**b**) Flexible arm assembly (METRx^®^). (**c**) Tubular retractor of the SYNCHA^®^ system. The black arrow indicates the ball link. (**d**) Scope attachment of the SYNCHA^®^ system. The black arrow indicates the nozzle for air suction, while the red arrow indicates the nozzle for the irrigation of the endoscope surface. (**e**) Variation of curved Kerrison rongeur. (**f**) Curved high-speed drill (Midas Rex^®^).

**Figure 2 medicina-58-00384-f002:**
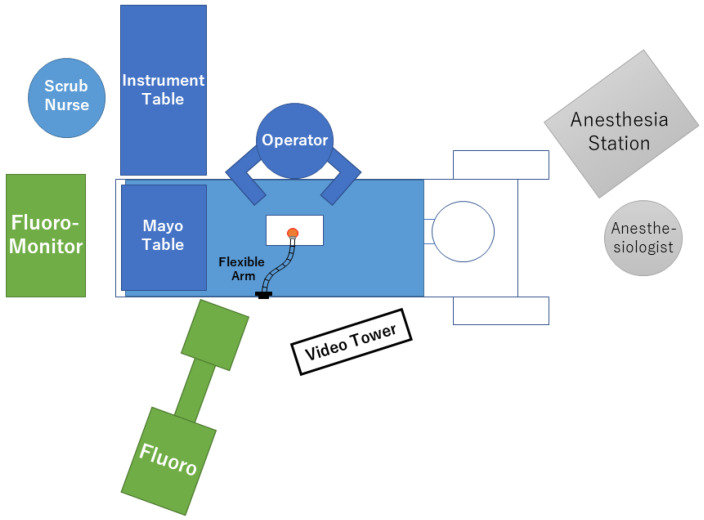
Schematic presentation of the room setup.

**Figure 3 medicina-58-00384-f003:**
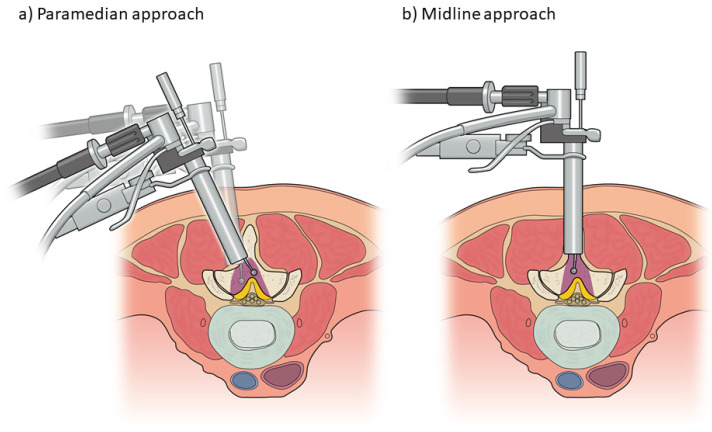
Schematic presentation of the (**a**) paramedian approach and (**b**) midline approach in microendoscopic lumbar decompression.

**Figure 4 medicina-58-00384-f004:**
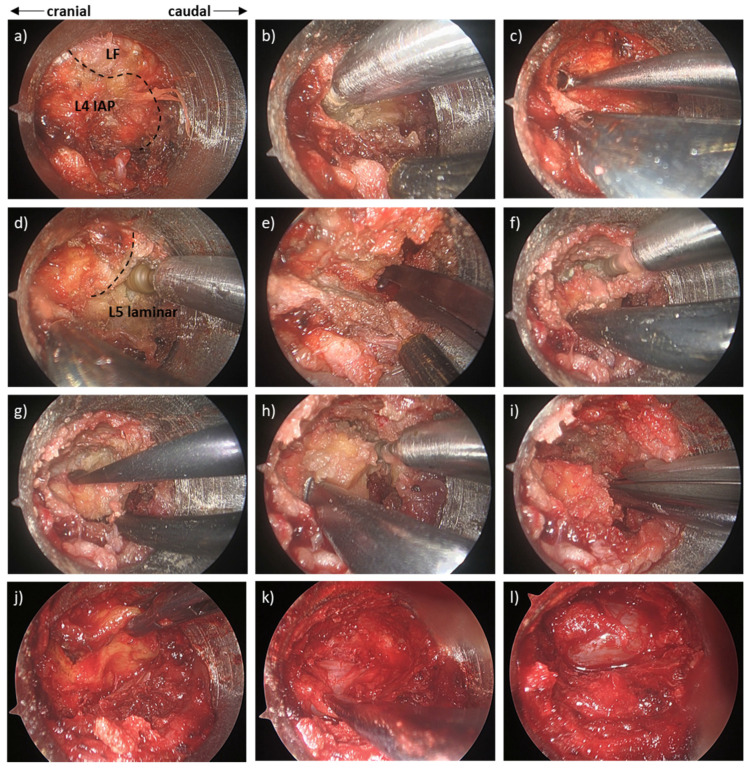
Intraoperative microendoscopic photograph during microendoscopic decompression for L4/5 lumbar spinal stenosis using a paramedian approach. (**a**) Before laminotomy (LF; ligamentum flavum, IAP; inferior articular process). (**b**) Drilling of the caudal side of the L4 lamina and medial part of the L4 IAP. (**c**) Detachment of the LF from the L4 lamina using a curved curette. (**d**) Drilling of the cranial side of the L5 lamina. (**e**) Resection of the medial side of the L5 superior articular process (SAP) using Kerrison rongeurs. (**f**) Drilling of the base of the L4 spinous process and contralateral L4 lamina and IAP. (**g**) Detachment of the LF from the contralateral L4 IAP using a dissector. (**h**) Drilling of the base of the L5 spinous process and contralateral L5 lamina and SAP. (**i**) Resection of the contralateral medial side of the L5 SAP using Kerrison rongeurs. (**j**) Removal of the LF. (**k**) Additional resection of the remaining LF and medial facet with confirmation of adequate decompression of the contralateral side. (**l**) Complete decompression after additional resection of the ipsilateral remaining LF and medial facet.

**Figure 5 medicina-58-00384-f005:**
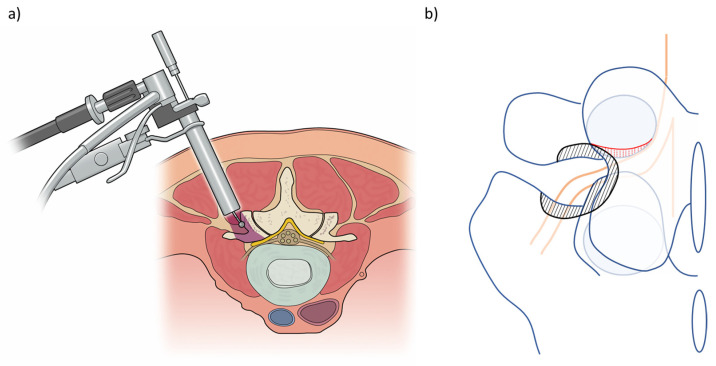
Schematic presentation of the (**a**) approach for foraminal stenosis and (**b**) example of the decompression area in a case of L5/S foraminal stenosis (black hatching area, bone removal area in usual decompression; red hatching area, additional pediculectomy in a case with up-down stenosis).

**Table 1 medicina-58-00384-t001:** Comparison of clinical outcomes between microendoscopic decompression and other surgical techniques.

First Author	Year	Comparison	Approach	No. of Patients	Follow-Up (Months)	Advantages of Microendoscopic Decompression	Disadvantages of Microendoscopic Decompression	Clinical Outcome in Microendoscopic Decompression Compared with the Opposite Arm	Complication in Microendoscopic Decompression Compared with the Opposite Arm	Ref. No.
Khoo	2002	vs. open	Paramedian	25 vs. open 25	12	Less blood loss Shorter hospital stay Less use of narcotics	-	Similar changes in symptom	Dural tear 16% vs. 8% Additional fusion surgery 0% vs. 12% Transfusion 0% vs. 8%	[33]
Ikuta	2005	vs. microsurgery	Paramedian	47 (DS 14) vs. micro 29 (DS 9)	22	Less blood loss Shorter hospital stay Less use of analgesics	Higher complication rate	Similar improvement in JOA score and VAS for low back pain and leg pain	Dural tear 8.5% vs. 6.8% Facet fracture 6.4% vs. 3.4% Transient neuralgia 14.9% vs. 3.4%	[20]
Rahman	2008	vs. open	Paramedian	38 vs. open 88	1	Shorter operating time Less blood loss Shorter hospital stay Lower complication rate	-	N/A	Dead 0% vs. 1.1% Wound exploration 0% vs. 2.3% Dural tear 2.6% vs. 2.3% Synovial cyst 2.6% vs. 1.1% Infection 2.6% vs. 3.4%	[49]
Fujimoto	2015	vs. microsurgery	Paramedian	21 vs. micro 20 (Including Myerding Grade1 DS)	24	Shorter operating time Less blood loss Shorter hospital stay Lower CRP Less dose of NSAIDs	-	Similar improvement in JOA score and VAS for leg pain	Transient neuralgia 15% vs. 4.8% Disturbance of wound healing 10% vs. 0%	[50]
Yagi	2009	vs. open	Midline	20 vs. open 21	12	Less blood loss Shorter hospital stay Lower CPK Less atrophy of PVM	-	Less VAS for low back pain Similar improvement in JOA score		[22]
Fukushi	2015	vs. spinous process splitting laminectomy	Midline	58 (DS 13) vs. open 39 (DS 8)	42 (>6)	Lower CRP	-	Similar improvement in JOABPEQ, SF-36, VAS Similar patient satisfaction scores	Superficial infection 3.4% vs. 0%	[51]
Hayashi	2018	vs. fusion (CBT-PLIF)	Paramedian	30 vs. fusion 20 (All patients had DS)	42 (>24)	Less blood loss Lower CRP Less dose of NSAIDs	-	Similar improvement in JOA score, and VAS for low back pain, leg pain, and leg numbness	Re-op 16% vs. 15% Neurological deficit 3.3% vs. 5.0% Epidural hematoma 0% vs. 5.0% Dural tear 6.7% vs. 0%	[56]
Aihara	2018	vs. fusion	Paramedian	25 vs. fusion 16 (All patients had DS)	60	Shorter operating time Less blood loss Shorter hospital stay	-	Greater improvement in the social function domain in JOABPEQ	Re-op 12% vs. 12.5%	[14]
Kimura	2019	vs. fusion (conventional PLIF)	Midline	37 vs. fusion 79 (Including Myerding Grade1 DS)	60	Shorter operating time Less blood loss	-	Similar improvement in JOA score, JOABPEQ, ZCQ, and VAS for low back pain, leg pain, leg numbness	Dural tear 2.7% vs. 1.3% Superficial infection 0% vs. 2.5% Pulmonary embolism 2.7% vs. 0% Re-op 7.1% vs. 8.0%	[57]
Wu	2020	vs. full endoscopic decompression	Paramedian	82 vs. 52	20	Shorter operating time	Longer skin incision	Similar improvement in VAS for leg pain Higher VAS for low back pain and ODI only at 1 week, and those are similar at 6 months and final follow-up	Total 3.85% vs. 3.66% Dural tear 2.4% vs. 1.9% Urinary retention 1.2% vs. 0% Dysesthesia 0% vs. 1.9%	[52]
Iwai	2020	vs. full endoscopic decompression	Paramedian	60 vs. 54 (Including Myerding Grade1 DS)	3	Shorter operating time	Longer hospital stay	Similar improvement in NRS	Dural tear 5.6% vs. 1.8% Hematoma 3.3% vs. 13.0%	[53]
Ito	2021	vs. full endoscopic decompression (biportal)	Paramedian	139 vs. 42	6	-	-	Similar improvement in VAS for low back pain and leg pain, ODI, and EQ5D	Dural tear 5.8% vs. 4.7% Hematoma 3.6% vs. 0% Re-op 1.4% vs. 0%	[54]
Aygun	2021	vs. full endoscopic decompression (biportal)	Paramedian	77 vs. 77 (Randomized controlled trial)	24	-	Longer hospital stay Longer operating time More blood loss	Less improvement in ODI, ZCQ Lower results in Modified MacNab criteria	N/A	[55]

CRP, C Reactive Protein; CBT, cortical bone trajectory; DS, degenerative spondylolisthesis; EQ5D, EuroQol 5-Dimension questionnaire; JOA score, Japanese Orthopaedic Association score; JOABPEQ, Japanese Orthopaedic Association Back Pain Evaluation Questionnaire; NSAIDs, Non-Steroidal Anti-Inflammatory Drugs; NRS, Numerical Rating Scale; ODI, Oswestry Disability Index; PLIF, Posterior Lumbar Interbody Fusion; PVM, Paravertebral Muscle; SF36, 36-Item Short Form Survey; VAS, Visual Analogue Scale; ZCQ, Zurich Claudication Questionnaire.

**Table 2 medicina-58-00384-t002:** Clinical results of microendoscopic decompression for lumbar foraminal stenosis.

Author	Year	No. of Patients	Level	Follow-Up (Months)	Clinical Outcome	Complication/Revision Surgery	Ref. No.
Matsumoto	2006	3	L5/S	31	JOA RR 42.7%	None	[28]
Zhou	2009	5	L5/S	19.7	Improvement of VAS (10 cm); 5.9	None	[60]
Matsumoto	2010	2 (microendoscopic 19, surgical loupe or microscopic 9)	L5/S	32.5	JOA RR 68.5% (No significant difference between surgical approaches)	Intraoperative blood loss exceeded 100 mL in 4 cases Revision surgery in 4 cases	[61]
Yamada	2012	32	L5/S	37.4	JOA RR 60.1% Improvement of VAS (100 mm); 68.2 in leg pain, 31.8 in low back pain, 39.7 in leg numbness	Painful dysesthesia in 1 case Recurrence of symptom in 4 cases Revision surgery in 2 cases	[24]
Yoshimoto	2019	20	L5/S in 16 cases L5/6 in 2 cases L4/5 in 2 cases	66.3	JOA RR 63.9%	Revision surgery in 5 cases	[25]
Murata	2020	78	L5/s	24	JOA RR 56.0% Improvement of VAS (100 mm); 49.0 in leg pain, 29.8 in low back pain,	Painful dysesthesia in 5 cases	[27]

## Data Availability

Not applicable.

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
