# Peer review of "Microendoscopic Lumbar Posterior Decompression Surgery for Lumbar Spinal Stenosis: Literature Review"

_medicina, 2022, doi:10.3390/medicina58030384_

Round 1

Reviewer 1 Report

Thank you for submitting interesting paper.

The manuscript was well written and scientific. However, I have some recommendations for your paper.

  1. Please add a figure for foraminal stenosis as shown in Figure 2.
  2. It would be good if a review about microendoscopic discectomy would also be included.
  3. Please add the intraoperative setting photo and microscopic view photo of microendoscopic decompression.

Thank you!

Author Response

Reviewer1.

Thank you for submitting interesting paper.

The manuscript was well written and scientific. However, I have some recommendations for your paper.

Thank you very much for your careful review. We appreciate your positive comments. Please see our point-by-point responses below.

  1. Please add a figure for foraminal stenosis as shown in Figure 2.

Thank you for this helpful suggestion. We have added images related to foraminal stenosis in Figure 3.

  1. It would be good if a review about microendoscopic discectomy would also be included.

Thank you for highlighting this issue. This article focuses on lumbar spinal stenosis in particular. Given that the word count is already over 7,500, we would appreciate if the reviewer would allow us to exclude content related to microendoscopic discectomy.

  1. Please add the intraoperative setting photo and microscopic view photo of microendoscopic decompression.

Thank you for your suggestion. Rather than a photograph, we have added an illustration of the intraoperative setup, which is now described in greater detail in the manuscript. In addition, we have included intraoperative microendoscopic photographs for decompression via the paramedian approach.

Reviewer 2 Report

This manuscript contains a detailed introduction of microendoscopic technique and comparison with various other decompression techniques for degenerative lumbar stenosis. Overall, the contents are well-organized and I think it will be great helpful to the readers working in the field of spine surgery. The reviewer’s additional thoughts about this paper are described below.

  1. It will be more helpful for the readers if the photos including the instruments (position and manipulation of the endoscope and other instruments) and operation view (through endoscope).
  2. In table 1, it is necessary to indicate the targeted diseases (simple stenosis, spondylolisthesis…) of each study comparing the performances of the techniques.
  3. To improve the reader’s readability, the contents of 7.5. are also recommended to be presented as the form of table.
  4. It is also recommended to organize the tables separately depending on the type of disease (simple stenosis, spondylolisthesis…).
  5. Please add a quotation for 'trumpet facetectomy' in ‘line 119’.
  6. I also think it will be worth it to reconfigure the manuscript according to the view of the readers who are beginner for spine surgery.

Author Response

Reviewer 2.

This manuscript contains a detailed introduction of microendoscopic technique and comparison with various other decompression techniques for degenerative lumbar stenosis. Overall, the contents are well-organized and I think it will be great helpful to the readers working in the field of spine surgery. The reviewer’s additional thoughts about this paper are described below.

Thank you very much for your careful review. We appreciate your positive comments. Please see our point-by-point responses below.

  1. It will be more helpful for the readers if the photos including the instruments (position and manipulation of the endoscope and other instruments) and operation view (through endoscope).

Thank you for this helpful suggestion. We have included intraoperative microendoscopic photographs for decompression via the paramedian approach.

  1. In table 1, it is necessary to indicate the targeted diseases (simple stenosis, spondylolisthesis…) of each study comparing the performances of the techniques.

Thank you for highlighting this issue. In many studies, patient characteristics were not uniform and included both simple stenosis and mild spondylolisthesis or were not reported in detail. However, we have added information regarding the rate of degenerative spondylolisthesis in Table 1.

  1. To improve the reader’s readability, the contents of 7.5. are also recommended to be presented as the form of table.

Thank you for this helpful suggestion. We have added Table 2 to improve readability.

  1. It is also recommended to organize the tables separately depending on the type of disease (simple stenosis, spondylolisthesis…).

Thank you for highlighting this issue. As described above, patient characteristics were not uniform and included both simple stenosis and mild spondylolisthesis or were not reported in detail in many studies. Therefore, it is difficult to separate the studies in this manner. Instead, we have organized the studies based on year of publication and operative methods for comparison.

  1. Please add a quotation for 'trumpet facetectomy' in ‘line 119’.

Thank you for this helpful suggestion. We have included quotation marks around “trumpet facetectomy.”

  1. I also think it will be worth it to reconfigure the manuscript according to the view of the readers who are beginner for spine surgery.

Thank you for this helpful recommendation. We have added a figure and a description regarding the room setup, as well as intraoperative microendoscopic photographs.

Reviewer 3 Report

This is a well written review providing a detailed scenario of micro endoscopic strategies and their impact on outcome compared to the standard microsurgical or open procedures. The topic is of interest and curiously underrated in the recent literature in terms of review with the latest paper published on Spine on 2016. The authors provide what is the core of a review treating this kind of technique: advantages and disadvantages. These are adequately analyzed and discussed with a table well structured which gives all the most significant information to the reader. The argumentations in favor or against the microendoscopic technique are well balanced and this is appreciated. I would support the paper for publication. 

Author Response

Reviewer 3.

This is a well written review providing a detailed scenario of micro endoscopic strategies and their impact on outcome compared to the standard microsurgical or open procedures. The topic is of interest and curiously underrated in the recent literature in terms of review with the latest paper published on Spine on 2016. The authors provide what is the core of a review treating this kind of technique: advantages and disadvantages. These are adequately analyzed and discussed with a table well structured which gives all the most significant information to the reader. The argumentations in favor or against the microendoscopic technique are well balanced and this is appreciated. I would support the paper for publication.

Thank you very much for your careful review, and we appreciate these positive comments.